# Probabilities of Causation: Adequate Size of Experimental and Observational Samples

**Ang Li**
Department of Computer Science
University of California Los Angeles
Los Angeles, CA 90095
angli@cs.ucla.edu

**Ruirui Mao**
Department of Statistics
University of Wisconsin Madison
Madison, Wisconsin 53705
rmao28@wisc.edu

**Judea Pearl**
Department of Computer Science
University of California Los Angeles
Los Angeles, CA 90095
judea@cs.ucla.edu

## Abstract

The probabilities of causation are commonly used to solve decision-making problems. Tian and Pearl derived sharp bounds for the probability of necessity and sufficiency (PNS), the probability of sufficiency (PS), and the probability of necessity (PN) using experimental and observational data. The assumption is that one is in possession of a large enough sample to permit an accurate estimation of the experimental and observational distributions. In this study, we present a method for determining the sample size needed for such estimation, when a given confidence interval (CI) is specified. We further show by simulation that the proposed sample size delivered stable estimations of the bounds of PNS.

## 1 Introduction

The probabilities of causation are widely used in many areas of industry, marketing, and health science, to solve decision-making problems. For example, Li and Pearl [6, 8] proposed the "benefit function", a linear combination of the probabilities of causation, which is the payoff/cost associated with selecting an individual with given characteristics to identify a set of individuals who are most likely to exhibit a desired mode of behavior. Mueller and Pearl [11] demonstrated, for example, that the probabilities of causation should be considered in personalized decision-making. Li et al. [7] showed that the probabilities of causation can improve the accuracy of machine learning algorithms.

Pearl [14] first used the structural causal model (SCM) to defined three binary probabilities of causation (i.e., PNS, PN, and PS) [3, 4, 15]. Tian and Pearl [16] then used experimental and observational data to bound those three probabilities of causation. Li and Pearl [8, 10] established formal proof of those bounds. Mueller, Li, and Pearl [12] proposed narrowing the bounds of PNS using covariate information and the causal structure. Dawid et al. [2] also proposed using covariate information to narrow the bounds of PN. Li and Pearl [5] recently established the theoretical bounds of nonbinary probabilities of causation.

All the abovementioned works are asymptotic (i.e., assuming a adequately large sample size to estimate the experimental and observational distributions). The proposed results in those works are relationships between the experimental and observational distributions and the probabilities of causation. However, the adequate sample size for obtaining those probabilities of causation remains

36th Conference on Neural Information Processing Systems (NeurIPS 2022).

unclear, thereby creating a barrier between the theoretical results and the real-world applications. Consider the following motivating example: a mobile carrier that wants to identify customers who are likely to discontinue their services within the next quarter based on customer characteristics (company management has access to user data, such as income, age, usage, and monthly payments). The carrier will then offer these customers a special renewal deal to dissuade them from discontinuing their services and to increase their service renewal rate. These offers provide considerable discounts to the customers, and the management prefers that these offers be made only to those customers who would continue to use the service if and only if they receive the offer. The manager decides to use Li and Pearl's unit selection model [8] but is unsure how many experimental and observational samples are required. Are 1000 experimental and 1000 observational samples adequate to bound the benefit function such that the error of the bounds are within 0.1?

We present an assessment of the "adequate" of the sample size in the sense of CI in this study. We would then be able to answer the question, "How many samples are adequate to estimate the probability of causation?" as "This amount of samples is adequate to obtain the bounds of the probability of causation in 95% CIs with margin of errors of 0.05." The probabilities of causation in most cases are not identifiable; therefore, the CIs are for the bounds of the probabilities of causation in such cases.

## 2 Preliminaries

We review the definitions for the three aspects of binary causation in this section, as defined in [14]. We use the language of counterfactuals in SCM, as defined in [3, 4]. We use $Y_x = y$ to denote the counterfactual sentence "Variable $Y$ would have the value $y$, had $X$ been $x$". For the rest of the paper, we use $y_x$ to denote the event $Y_x = y$, $y_{x'}$ to denote the event $Y_{x'} = y$, $y'_x$ to denote the event $Y_x = y'$, and $y'_{x'}$ to denote the event $Y_{x'} = y'$. We assume that experimental distribution will be summarized in the form of the causal effects such as $P(y_x)$ and observational distribution will be summarized in the form of the joint probability function such as $P(x, y)$. If neither variable is specified, variable $X$ represents treatment and variable $Y$ represents effect.

The following are three prominent probabilities of causation:

**Definition 1** (Probability of necessity (PN)). *Let $X$ and $Y$ be two binary variables in a causal model $M$, let $x$ and $y$ stand for the propositions $X = true$ and $Y = true$, respectively, and $x'$ and $y'$ for their complements. The probability of necessity is defined as the expression [14]*

$$PN \triangleq P(Y_{x'} = false | X = true, Y = true) \triangleq P(y'_{x'} | x, y)$$

**Definition 2** (Probability of sufficiency (PS)). *[14]*

$$PS \triangleq P(y_x | y', x')$$

**Definition 3** (Probability of necessity and sufficiency (PNS)). *[14]*

$$PNS \triangleq P(y_x, y'_{x'})$$

PNS denotes for the probability that $y$ would respond to $x$ both ways, and therefore measures both the sufficiency and necessity of $x$ to produce $y$.

Tian and Pearl [16] used Balke's program [1] to provide tight bounds for PNS, PN, and PS.

PNS has the following tight bounds:

$$\max \left\{ \begin{array}{c} 0, \\ P(y_x) - P(y_{x'}), \\ P(y) - P(y_{x'}), \\ P(y_x) - P(y) \end{array} \right\} \leq PNS \leq \min \left\{ \begin{array}{c} P(y_x), \\ P(y'_{x'}), \\ P(x, y) + P(x', y'), \\ P(y_x) - P(y_{x'}) + P(x, y') + P(x', y) \end{array} \right\} \quad (1)$$

Note that we omitted the bounds of PN and PS because this study focuses primarily on the adequate sample size for estimating the bounds of PNS, it is simple to extend to other probabilities of causation.

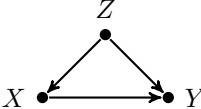

Figure 1: The Causal Model, where $X$ is a binary treatment, $Y$ is a binary effect, and $Z$ is a set of $20$ independent binary confounders.

## 3   Main Results

The bounds of PNS are the linear combination of the experimental distributions $P(y_x), P(y_{x'})$ and the observational distributions $P(x, y), P(x, y'), P(x', y), P(x', y')$ from Equation 1. Therefore, if we can obtain the CI of each of these distributions, then we can obtain the CIs of the bounds of the PNS. Let $R$ be a random variable such that $R = 1$ if the event $y_x$ occurs and $R = 0$ if the event $y'_x$ occurs, then it is clear that $R \sim Bernoulli(P(y_x))$. Therefore, if we use the frequentist to estimate the experimental and observational distributions, we have the following theorem and corollary (the detailed proof are in the appendix):

**Theorem 4.** *Given $m$ experimental samples and $n$ observational samples, if the frequentist is used to estimate the experimental and observational distributions, then the margin error of the bounds of PNS in $1 - \alpha$ confidence interval is at most $z_{1-\alpha/2}(\sqrt{\frac{1}{m}} + \sqrt{\frac{1}{n}})$, where $z_{1-\alpha/2}$ can be found on z-table of standard normal distribution.*

**Corollary 5.** *If the frequentist is used to estimate the experimental and observational distribution, to obtain the at most $0.05$ margin error of the bounds of PNS in $95\%$ confidence interval, we need $m$ experimental samples and $n$ observational samples, where $(\sqrt{\frac{1}{m}} + \sqrt{\frac{1}{n}}) \leq 5/196$. More specifically, if $m = n$, $6147$ experimental and $6147$ observational samples are adequate to obtain the at most $0.05$ margin error of the bounds of PNS in $95\%$ confidence interval.*

This amount of samples ensures that the margin errors of the bounds of PNS in $95\%$ CI are no more than $0.05$. However, in practice, we usually do not need this amount of samples because there is only one term (i.e., $P(y_x) - P(y_{x'}) + P(x, y') + P(x', y)$) in the PNS bounds of Equation 1, which consists of four distributions. The terms such as $P(y_x)$ only require $385$ experimental and $385$ observational samples to obtain the at most $0.05$ margin error of the $95\%$ CI, and the terms such as $P(y_x) - P(y_{x'})$ only require $1537$ experimental and $1537$ observational samples to obtain the at most $0.05$ margin error of the $95\%$ CI. We will illustrate the real errors of the estimations in simulated studies in the following section.

## 4   Simulation Results

Here, we present simulated studies to show that the proposed number of experimental and observational samples are adequate to obtain the desired margin errors of the bounds of PNS using two SCMs.

### 4.1   Causal Model

To estimate the margin errors of the bounds, we must first understand the data generation process to have true experimental and observational distributions. The two models we are using are shown in Figure 1 (two models have the same causal graph, but with different coefficients in SCMs; the generation method of the models is in the appendix), where $X$ is a binary treatment, $Y$ is a binary effect, and $Z$ is a set of 20 independent confounders (say $Z_1, ..., Z_{20}$). The structural equations are as follow (for simplicity reason, we let $x = 1, x' = 0$, and $y = 1, y' = 0$):

$$Z_i = U_{Z_i} \text{ for } i \in \{1, ..., 20\},$$

$$X = f_X(M_X, U_X) = \left\{ \begin{array}{ll} 1 & \text{if } M_X + U_X > 0.5, \\ 0 & \text{otherwize,} \end{array} \right\}$$

$$Y = f_Y(X, M_Y, U_Y) = \left\{ \begin{array}{ll} 1 & \text{if } 0 < CX + M_Y + U_Y < 1 \text{ or } 1 < CX + M_Y + U_Y < 2, \\ 0 & \text{otherwize,} \end{array} \right\}$$

where, $U_{Z_i}, U_X, U_Y$ are binary exogenous variables with Bernoulli distributions,

C is a constant, and $M_X, M_Y$ are linear combinatations of $Z_1, ..., Z_{20}$.

The value of $C, M_X, M_Y$ and the distributions of $U_X, U_Y, U_{Z_1}, ..., U_{Z_{20}}$ for the two models are provided in the appendix.

## 4.2 Informer Data

Based on the model from the previous section, individuals are determined by 22 binary exogenous variables. From the viewpoint of the informer, we must know the actual experimental (i.e., $P(y_x), P(y_{x'})$) and observational distributions (i.e., $P(x, y), P(x, y'), P(x', y)$) to compute the true PNS bounds for the comparison purpose. Since the structural equations and the exogenous variable distributions are explicit, the experimental and observational distributions are as follows (we illustrated two terms, see the appendix for full details):

$$P(y_x) = \sum_{U_X, U_Y, U_{Z_1}, ..., U_{Z_{20}}} P(u_X)P(u_Y)P(u_{Z_1}) \times ... \times P(u_{Z_{20}})f_Y(1, M_Y, u_Y),$$

$$P(x, y) = \sum_{U_X, U_Y, U_{Z_1}, ..., U_{Z_{20}}} P(u_X)P(u_Y)P(u_{Z_1}) \times ... \times P(u_{Z_{20}}) \times$$
$$\times f_X(M_X, u_X)f_Y(f_X(M_X, u_X), M_Y, u_Y),$$

We can then obtain the informer view of the bounds of PNS using Equation 1 and the above observational and experimental distributions.

## 4.3 Experimental Sample

Here is how we prepared the finite experimental samples. Again, an individual is determined by $U_X$, $U_Y$, and $U_{Z_1}, ..., U_{Z_{20}}$. Therefore, we first generated $(U_X, U_Y, U_{Z_1}, ..., U_{Z_{20}})$ at random using their distributions; then we generated $X$ at random using $Bernoulli(0.5)$; the value of $Y$ is then $f_Y(X, M_Y, U_Y)$. We then collect a experimental sample $(X, Y)$. $P(y_x)$ is then estimated as $\hat{P}(y_x) = \frac{\text{the number of experimental samples (1,1)}}{\text{the number of experimental samples (1,Y)}}$. $P(y_{x'})$ is then estimated as $\hat{P}(y_{x'}) = \frac{\text{the number of experimental samples (0,1)}}{\text{the number of experimental samples (0,Y)}}$.

## 4.4 Observational Sample

Similarly to experimental sample, we first generated $(U_X, U_Y, U_{Z_1}, ..., U_{Z_{20}})$ at random using their distributions; the value of $X$ is then $f_X(M_X, U_X)$; the value of $Y$ is then $f_Y(X, M_Y, U_Y)$. Then, we collect a observational sample $(X, Y)$. $P(x, y), P(x, y'), P(x', y)$ estimates are then $\hat{P}(x, y) = \frac{\text{the number of observational samples (1,1)}}{\text{the number of total observational samples}}$, $\hat{P}(x, y') = \frac{\text{the number of observational samples (1,0)}}{\text{the number of total observational samples}}$, and $\hat{P}(x', y) = \frac{\text{the number of observational samples (0,1)}}{\text{the number of total observational samples}}$.

## 4.5 Simulation Results

For the two SCMs, we randomly generated $385, 1537$, and $6147$ experimental and $385, 1537$, and $6147$ observational samples respectively for 1000 times as described in the previous sections. The true bounds of the PNS obtained from true distributions were then compared to the estimated bounds obtained from finite experimental and observational samples. Figures 2 and 3 show the comparison results. We see that when the sample size is 385 or even 1537, the bounds are still in a relatively large unstable situation, with some of the estimated lower bounds being even larger than the true PNS

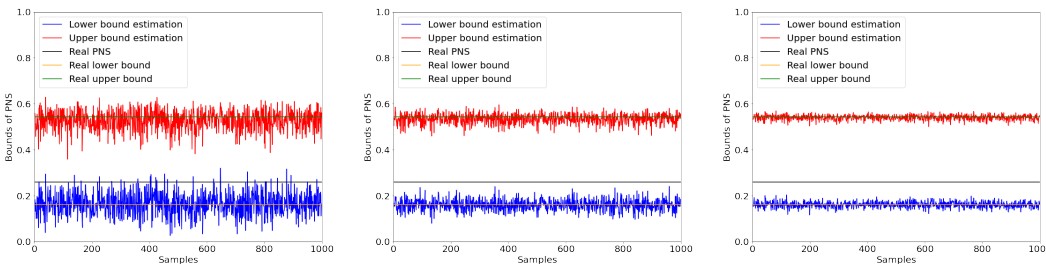

(a) 385 experimental and 385 observational samples.

(b) 1537 experimental and 1537 observational samples.

(c) 6147 experimental and 6147 observational samples.

Figure 2: Estimation of the bounds of PNS for the first model using different size of samples.

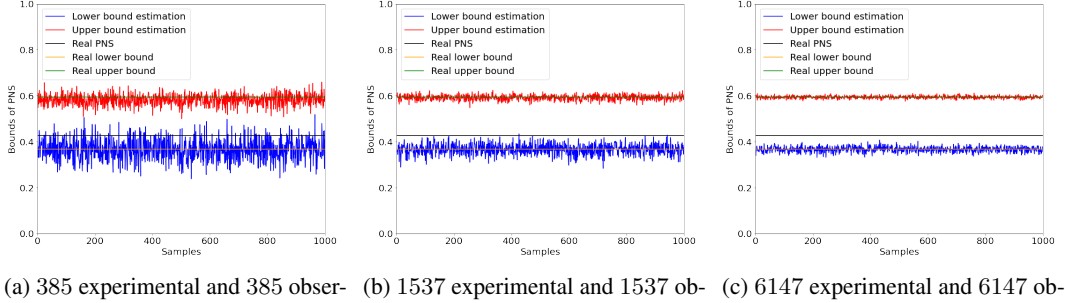

(a) 385 experimental and 385 observational samples.

(b) 1537 experimental and 1537 observational samples.

(c) 6147 experimental and 6147 observational samples.

Figure 3: Estimation of the bounds of PNS for the second model using different size of samples.

value. When the sample size reached $6147$, the estimations appear very stable and always include the true PNS value. We also plot the sample size v.s. the average error of estimations (i.e., |estimated bounds − true bounds|/1000) as shown in Figure 4. The average errors improved significantly before $1500$ sample sizes, therefore, $1537$ should be a good sample size to use in practice even though the adequate sample size from the theorem is $6147$. Besides, starting at roughly $300$ sample size, the average errors are less than the expected margin error of $0.05$. This is because the proposed theorem is of adequate size and considers the worst-case scenario.

## 5   Discussion

We demonstrated how to evaluate the precision of the probability of causation estimation using PNS. It is simple to extend to nonbinary probabilities of causation as defined in [5]. In such cases, we can still define a random variable, $R$, that obeys the Bernoulli distribution and whose parameter is the experimental distribution $P(y_x)$. This work can be extended to unit selection problems [6, 8], because the unit selection problems are linear combination of the probabilities of causation. This work can be extended to estimate experimental distributions [9, 13] using observational data.

## 6   Conclusion

We showed how to assess the accuracy of a PNS bounds estimation given by a finite number of experimental and observational samples, as well as how many experimental and observational samples are adequate to achieve a certain accuracy level of a PNS bounds estimation. Both theoretical and experimental results are provided. The proposed method can also be extended to general probabilities of causation.

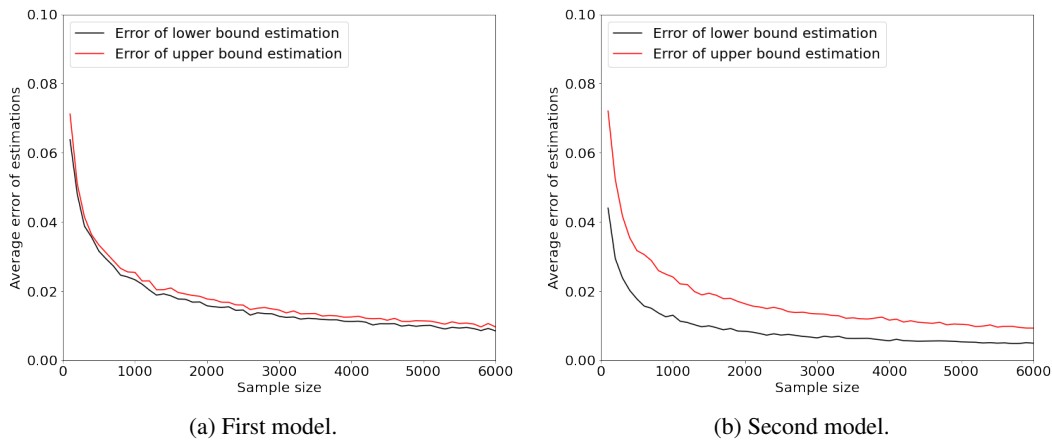

(a) First model.            (b) Second model.

Figure 4: Average error of estimations using different size of data.

## Acknowledgements

This research was supported in parts by grants from the National Science Foundation [#IIS-2106908 and #IIS-2231798], Office of Naval Research [#N00014-21-1-2351], and Toyota Research Institute of North America [#PO-000897].

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

# A Appendix

## A.1 Proof of Theorem 4

*Proof.* Let $R$ be a random variable, such that $R = 1$ if the event $y_x$ occurs and $R = 0$ if the event $y'_x$ occrus, then $R \sim Bernoulli(P(y_x))$.

Let $R = (R_1, ..., R_m)$ be $m$ random experimental samples of $R$, and $S_m = \sum_{i=1}^{m} R_i$.

Therefore, $S_m \sim binomial(m, P(y_x))$.

By Central limit theorem, we have,

$\frac{S_m - mp}{\sqrt{mP(y_x)(1-P(y_x))}} \to N(0, 1)$ as $m \to \infty$.

We also have $\hat{P}(y_x) = S_m / m \xrightarrow{P} P(y_x)$,

Thus, we have,

$\sqrt{\frac{P(y_x)(1-P(y_x))}{\hat{P}(y_x)(1-\hat{P}(y_x))}} \xrightarrow{P} 1$ and $\frac{\hat{P}(y_x) - P(y_x)}{\sqrt{P(y_x)(1-P(y_x))/m}} \xrightarrow{L} N(0, 1)$.

Therefore, we have,

$\frac{\hat{P}(y_x) - P(y_x)}{\sqrt{\hat{P}(y_x)(1-\hat{P}(y_x))/m}} \xrightarrow{L} N(0, 1)$.

We know that $P(|\frac{\hat{P}(y_x) - P(y_x)}{\sqrt{\hat{P}(y_x)(1-\hat{P}(y_x))/m}}| \le z_{1-\alpha/2}) = 1 - \alpha$,

we have the confidence interval of $\hat{P}(y_x)$ is

$[\hat{P}(y_x) - z_{1-\alpha/2} \cdot \sqrt{\hat{P}(y_x)(1 - \hat{P}(y_x))/m}, \hat{P}(y_x) + z_{1-\alpha/2} \cdot \sqrt{\hat{P}(y_x)(1 - \hat{P}(y_x))/m}]$.

The margin of error of $\hat{P}(y_x)$ is $z_{1-\alpha/2} \cdot \sqrt{\hat{P}(y_x)(1 - \hat{P}(y_x))/m}$, denoted as

$err(\hat{P}(y_x)) = z_{1-\alpha/2} \cdot \sqrt{\hat{P}(y_x)(1 - \hat{P}(y_x))/m}$.

We then have,

$err(\hat{P}(y_x)) = z_{1-\alpha/2} \cdot \sqrt{\hat{P}(y_x)(1 - \hat{P}(y_x))/m} \le z_{1-\alpha/2} \cdot \sqrt{\frac{1}{4m}} = \frac{z_{1-\alpha/2}}{2} \sqrt{\frac{1}{m}}$.

Similarly, we have,

$err(\hat{P}(y_{x'})) \le \frac{z_{1-\alpha/2}}{2} \sqrt{\frac{1}{m}}$,

$err(\hat{P}(y)) \le \frac{z_{1-\alpha/2}}{2} \sqrt{\frac{1}{n}}$,

$err(\hat{P}(x, y)) \le \frac{z_{1-\alpha/2}}{2} \sqrt{\frac{1}{n}}$,

$err(\hat{P}(x', y)) \le \frac{z_{1-\alpha/2}}{2} \sqrt{\frac{1}{n}}$,

$err(\hat{P}(x, y')) \le \frac{z_{1-\alpha/2}}{2} \sqrt{\frac{1}{n}}$,

$err(\hat{P}(x', y')) \le \frac{z_{1-\alpha/2}}{2} \sqrt{\frac{1}{n}}$.

We also have the fact that $err(A + B) \le err(A) + err(B)$ and $err(A - B) \le err(A) + err(B)$,

plug into Equation 1, we obtain that $err(\text{bounds of PNS}) \le z_{1-\alpha/2}(\sqrt{\frac{1}{m}} + \sqrt{\frac{1}{n}})$. $\qquad\square$

## A.2 Proof of Corollary 5

*Proof.* Let $\alpha = 0.05$.

We need $z_{0.975}(\sqrt{\frac{1}{m}} + \sqrt{\frac{1}{n}}) \le 0.05$.

Simply plug $z_{0.975} = 1.96$ in, we have,

$(\sqrt{\frac{1}{m}} + \sqrt{\frac{1}{n}}) \le 5/196$.

And if $m = n$, we have,

$\sqrt{\frac{1}{m}} \le 5/392$,

Therefore, $m \ge 6146.56$. $\qquad\square$

### A.3 Two Causal Models

First, $M_X$ and $M_Y$ are both linear combinatation of $Z_1, ..., Z_{20}$. So for each model, we need to generate 20 coefficients for $M_X$, 20 coefficients for $M_Y$, a constant $C$, and 22 Bernoulli distribution parameters for $U_X, U_Y, U_{Z_1}, ..., U_{Z_{20}}$. We generated the coeefficients for $M_X$, $M_Y$ and $C$ uniformly from $[-1, 1]$, and generated the Bernoulli distribution parameters unoformly from $[0, 1]$. The detailed two models are as follow:

### A.3.1 Model 1

$$Z_i = U_{Z_i} \text{ for } i \in \{1, ..., 20\},$$

$$X = f_X(M_X, U_X) = \left\{ \begin{array}{ll} 1 & \text{if } M_X + U_X > 0.5, \\ 0 & \text{otherwize}, \end{array} \right\}$$

$$Y = f_Y(X, M_Y, U_Y) = \left\{ \begin{array}{ll} 1 & \text{if } 0 < CX + M_Y + U_Y < 1 \text{ or } 1 < CX + M_Y + U_Y < 2, \\ 0 & \text{otherwize}, \end{array} \right\}$$

where, $U_{Z_i}, U_X, U_Y$ are binary exogenous variables with Bernoulli distributions.$s.t.$,

$U_{Z_1} \sim \text{Bernoulli}(0.352913861526), U_{Z_2} \sim \text{Bernoulli}(0.460995855543),$

$U_{Z_3} \sim \text{Bernoulli}(0.331702473392), U_{Z_4} \sim \text{Bernoulli}(0.885505026779),$

$U_{Z_5} \sim \text{Bernoulli}(0.017026872706), U_{Z_6} \sim \text{Bernoulli}(0.380772701708),$

$U_{Z_7} \sim \text{Bernoulli}(0.028092602705), U_{Z_8} \sim \text{Bernoulli}(0.220819399962),$

$U_{Z_9} \sim \text{Bernoulli}(0.617742227477), U_{Z_{10}} \sim \text{Bernoulli}(0.981975046713),$

$U_{Z_{11}} \sim \text{Bernoulli}(0.142042291381), U_{Z_{12}} \sim \text{Bernoulli}(0.833602592350),$

$U_{Z_{13}} \sim \text{Bernoulli}(0.882938907115), U_{Z_{14}} \sim \text{Bernoulli}(0.542143191999),$

$U_{Z_{15}} \sim \text{Bernoulli}(0.085023436884), U_{Z_{16}} \sim \text{Bernoulli}(0.645357252864),$

$U_{Z_{17}} \sim \text{Bernoulli}(0.863787135134), U_{Z_{18}} \sim \text{Bernoulli}(0.460539711624),$

$U_{Z_{19}} \sim \text{Bernoulli}(0.314014079207), U_{Z_{20}} \sim \text{Bernoulli}(0.685879388218),$

$U_X \sim \text{Bernoulli}(0.601680857267), U_Y \sim \text{Bernoulli}(0.497668975278),$

$C = -0.77953605542,$

$$M_X = [Z_1 \ Z_2 \ ... \ Z_{20}] \times \begin{bmatrix} 0.259223510143 \\ -0.658140989167 \\ -0.75025831768 \\ 0.162906462426 \\ 0.652023463285 \\ -0.0892939586541 \\ 0.421469107769 \\ -0.443129684766 \\ 0.802624388789 \\ -0.225740978499 \\ 0.716621631717 \\ 0.0650682260309 \\ -0.220690334026 \\ 0.156355773665 \\ -0.50693672491 \\ -0.707060278115 \\ 0.418812816935 \\ -0.0822118703986 \\ 0.769299853833 \\ -0.511585391002 \end{bmatrix}, M_Y = [Z_1 \ Z_2 \ ... \ Z_{20}] \times \begin{bmatrix} -0.792867111918 \\ 0.759967136147 \\ 0.55437722369 \\ 0.503970540409 \\ -0.527187144651 \\ 0.378619988091 \\ 0.269255196301 \\ 0.671597043594 \\ 0.396010142274 \\ 0.325228576643 \\ 0.657808327574 \\ 0.801655023993 \\ 0.0907679484097 \\ -0.0713852594543 \\ -0.0691046005285 \\ -0.222582013343 \\ -0.848408031595 \\ -0.584285069026 \\ -0.324874831799 \\ 0.625621583197 \end{bmatrix}$$

### A.3.2 Model 2

$$Z_i = U_{Z_i} \text{ for } i \in \{1, ..., 20\},$$

$$X = f_X(M_X, U_X) = \left\{ \begin{array}{ll} 1 & \text{if } M_X + U_X > 0.5, \\ 0 & \text{otherwize,} \end{array} \right\}$$

$$Y = f_Y(X, M_Y, U_Y) = \left\{ \begin{array}{ll} 1 & \text{if } 0 < CX + M_Y + U_Y < 1 \text{ or } 1 < CX + M_Y + U_Y < 2, \\ 0 & \text{otherwize,} \end{array} \right\}$$

where, $U_{Z_i}, U_X, U_Y$ are binary exogenous variables with Bernoulli distributions.*s.t.*,

$U_{Z_1} \sim \text{Bernoulli}(0.524110233482), U_{Z_2} \sim \text{Bernoulli}(0.689566064108),$

$U_{Z_3} \sim \text{Bernoulli}(0.180145428970), U_{Z_4} \sim \text{Bernoulli}(0.317153536644),$

$U_{Z_5} \sim \text{Bernoulli}(0.046268153873), U_{Z_6} \sim \text{Bernoulli}(0.340145244411),$

$U_{Z_7} \sim \text{Bernoulli}(0.100912238566), U_{Z_8} \sim \text{Bernoulli}(0.772038172066),$

$U_{Z_9} \sim \text{Bernoulli}(0.913108434869), U_{Z_{10}} \sim \text{Bernoulli}(0.364272299067),$

$U_{Z_{11}} \sim \text{Bernoulli}(0.063667554704), U_{Z_{12}} \sim \text{Bernoulli}(0.454839320009),$

$U_{Z_{13}} \sim \text{Bernoulli}(0.586687215140), U_{Z_{14}} \sim \text{Bernoulli}(0.018824647595),$

$U_{Z_{15}} \sim \text{Bernoulli}(0.871017316787), U_{Z_{16}} \sim \text{Bernoulli}(0.164966968157),$

$U_{Z_{17}} \sim \text{Bernoulli}(0.578925020078), U_{Z_{18}} \sim \text{Bernoulli}(0.983082980658),$

$U_{Z_{19}} \sim \text{Bernoulli}(0.018033993991), U_{Z_{20}} \sim \text{Bernoulli}(0.074629121266),$

$U_X \sim \text{Bernoulli}(0.29908139311), U_Y \sim \text{Bernoulli}(0.9226108109253),$

$C = 0.975140894243,$

$$M_X = [Z_1 \ Z_2 \ ... \ Z_{20}] \times \begin{bmatrix} 0.843870221861 \\ 0.178759296447 \\ -0.372349746729 \\ -0.950904544846 \\ -0.439457721339 \\ -0.725970103834 \\ -0.791203963585 \\ -0.843183562918 \\ -0.68422616618 \\ -0.782051030131 \\ -0.434420454146 \\ -0.445019418094 \\ 0.751698021555 \\ -0.185984172192 \\ 0.191948271392 \\ 0.401334543567 \\ 0.331387702568 \\ 0.522595634402 \\ -0.928734581669 \\ 0.203436441511 \end{bmatrix}, M_Y = [Z_1 \ Z_2 \ ... \ Z_{20}] \times \begin{bmatrix} -0.453251661832 \\ 0.424563325534 \\ 0.0924810605305 \\ 0.312680246141 \\ 0.7676961338 \\ 0.124337421843 \\ -0.435341306455 \\ 0.248957751703 \\ -0.161303883519 \\ -0.537653062121 \\ -0.222087991408 \\ 0.190167775134 \\ -0.788147770713 \\ -0.593030174012 \\ -0.308066297974 \\ 0.218776507777 \\ -0.751253645088 \\ -0.11151455376 \\ 0.785227235182 \\ -0.568046522383 \end{bmatrix}$$

## A.4 Informer Data

Detailed informer data can be obtained via the following equations:

$$
\begin{aligned}
P(y_x) &= \sum_{U_X, U_Y, U_{Z_1}, \ldots, U_{Z_{20}}} P(u_X)P(u_Y)P(u_{Z_1}) \times \ldots \times P(u_{Z_{20}})f_Y(1, M_Y, u_Y), \\
P(y_{x'}) &= \sum_{U_X, U_Y, U_{Z_1}, \ldots, U_{Z_{20}}} P(u_X)P(u_Y)P(u_{Z_1}) \times \ldots \times P(u_{Z_{20}})f_Y(0, M_Y, u_Y), \\
P(x, y) &= \sum_{U_X, U_Y, U_{Z_1}, \ldots, U_{Z_{20}}} P(u_X)P(u_Y)P(u_{Z_1}) \times \ldots \times P(u_{Z_{20}}) \times \\
&\quad \times f_X(M_X, u_X)f_Y(f_X(M_X, u_X), M_Y, u_Y), \\
P(x, y') &= \sum_{U_X, U_Y, U_{Z_1}, \ldots, U_{Z_{20}}} P(u_X)P(u_Y)P(u_{Z_1}) \times \ldots \times P(u_{Z_{20}}) \times \\
&\quad \times f_X(M_X, u_X)(1 - f_Y(f_X(M_X, u_X), M_Y, u_Y)), \\
P(x', y) &= \sum_{U_X, U_Y, U_{Z_1}, \ldots, U_{Z_{20}}} P(u_X)P(u_Y)P(u_{Z_1}) \times \ldots \times P(u_{Z_{20}}) \times \\
&\quad \times (1 - f_X(M_X, u_X))f_Y(f_X(M_X, u_X), M_Y, u_Y). \\
P(x', y') &= \sum_{U_X, U_Y, U_{Z_1}, \ldots, U_{Z_{20}}} P(u_X)P(u_Y)P(u_{Z_1}) \times \ldots \times P(u_{Z_{20}}) \times \\
&\quad \times (1 - f_X(M_X, u_X))(1 - f_Y(f_X(M_X, u_X), M_Y, u_Y)).
\end{aligned}
$$

