# OpenReview forum: "Probabilities of Causation: Adequate Size of Experimental and Observational Samples"
_NeurIPS.cc/2022/Workshop/nCSI — nCSI WS @ NeurIPS 2022 Poster_

### Official Review · Reviewer_SGk5 · 2022-10-15
**Paper review**

**Rating:** 2
**Confidence:** 3

**Review:**

The authors present a method to compute the sample size needed for estimating the bounds of PNS (probability of necessity and sufficiency) with a given confidence interval. The result of the paper is technically simple and straightforward. It leverages the facts that X and Y are binary variables and that the bounds for $PNS = P(y_x, {y'}_{x'})$ are linear combinations of the observational and experimental distributions. The idea simply relies on an indicator (binary) variable R for the events $y_x$ (R=1) and $y’_x$ (R=0), for both $X=x$ (true) and $X=x’$ (false). Despite the simplicity of the result, it is relevant for practical applications.

Simulations are conducted to illustrate the estimation of the bounds of PNS with a confidence of 95% in two different SCMs. The simulations were conducted 1000 times. The results show that the number of samples is adequate, even though it considers the worst-case scenario, and, therefore, smaller sample sizes could be sufficient for simpler models.


Minors:

1) ’The assumption is that one is in possession of a large enough sample to permit an accurate estimation of the experimental and observational distributions.’ — I wouldn’t say that this is an assumption of the theoretical bounds by Tian and Pearl. They proposed an estimator in terms of the real probability distributions, not of their estimates. Of course, if one needs to estimate these distributions in practical settings, large samples will lead to better estimates.

2) In Figure 2, the label for the X-axis is a bit confusing, since samples could be the sample size. I suggest using another label for it, for example, ‘number of simulation trials'.

3) In the appendix, line 231, there is a typo: occrus —> occurs

---

### Official Review · Reviewer_Uu7y · 2022-10-15

**Rating:** 2
**Confidence:** 2

**Review:**

The paper studies the sample size (of observational and experimental data needed to estimate the probability of causation, given a specified confidence interval. The authors derive the number of “adequate” sample sizes and evaluate their theoretical results empirically while estimating the PNS

$Pros$
-  The paper proposes a theorem describing the margin errors for the bounds of PNS given a confidence interval.
-  The theoretical results are tested on a synthetical data, confirming that when the sample size reaches the predicated number the estimated bounds appear more stable.
-  The authors provide also a discussion on the relationship between the sample size of experimental and observational data on the average error of estimates, giving guidelines on when the largest drop in the error occurs and  thus what would be a recommended minimal size of the data.

$Cons$
-  I would like the authors to comment on the impact of some design choices of the experiments on the obtained results, or the justification for the used models.  In particular, why the authors use only two models? Are those models meant to simulate some true data, or are purely synthetic (if, yes, how have they been generated – i.e. I can see the description in the appendix, but I am not sure what was the method for choosing the parameters of the distributions).  Similarly, why have the authors chosen to use 20 confounders?

In general I would also like to see how the work can be extended to real-world application and problems (something that authors mention in the Discussion, and which would be nice to see in the future).

---

### Meta-Review · Area_Chair_qXcg · 2022-10-17

**Recommendation:** 1
**Confidence:** 4

**Metareview:**

This paper considers the upper and lower bounds of the probability of necessity and sufficiency (PNS) derived by Tian and Pearl. The authors propose to compute CI’s for the estimation of the upper and lower bounds in order to use them to decide on how large the sample size needs to be to achieve a certain CI size. As the authors assume that the variables X and Y are binary, this is simply a trivial Binomial CI, which is further approximated with a normal distribution. As these types of CI’s are well-known in the literature and also common among practitioners, this contribution is not sufficiently interesting for this workshop. I therefore recommend to reject this submission.

---

### Decision · Program_Chairs · 2022-10-20

Accept (Poster)